# Automation protocol for high-efficiency and high-quality genomic DNA extraction from *Saccharomyces cerevisiae*

**Nina Alperovich**[1], **Benjamin M. Scott**[1,2¤], **David Ross**[1]*

**1** National Institute of Standards and Technology, Gaithersburg, MD, United States of America,
**2** Department of Chemistry and Biochemistry, University of Maryland, College Park, MD, United States of America

¤ Current address: Global Institute for Food Security, University of Saskatchewan, Saskatoon, SK, Canada
* david.ross@nist.gov.

**Data Availability Statement:** All relevant data are within the paper and its Supporting Information files.

**Funding:** The author(s) received no specific funding for this work.

## Abstract

Although many protocols have been previously developed for genomic DNA (gDNA) extraction from *S. cerevisiae*, to take advantage of recent advances in laboratory automation and DNA-barcode sequencing, there is a need for automated methods that can provide high-quality gDNA at high efficiency. Here, we describe and demonstrate a fully automated protocol that includes five basic steps: cell wall and RNA digestion, cell lysis, DNA binding to magnetic beads, washing with ethanol, and elution. Our protocol avoids the use of hazardous reagents (e.g., phenol, chloroform), glass beads for mechanical cell disruption, or incubation of samples at 100°C (i.e., boiling). We show that our protocol can extract gDNA with high efficiency both from cells grown in liquid culture and from colonies grown on agar plates. We also show results from gel electrophoresis that demonstrate that the resulting gDNA is of high quality.

## Introduction

Baker's yeast (*Saccharomyces cerevisiae*) is one of the most commonly used organisms for synthetic biology. Unlike many eukaryotes, *S. cerevisiae* prefers to repair DNA by homologous recombination, which enables the incorporation of custom genetic sequences into the genome with as little as ~50 bp of homology flanking a sequence [1, 2]. This feature has been used for decades to insert custom sequences into the genome for a wide array of biotechnology applications, including the assembly of whole genomes from kilobase-sized fragments [3], and the assembly of custom plasmid sequences from linear DNA [4]. Although plasmids are flexible tools for assembling transcriptional units [5], there is greater variability in gene expression from plasmids versus from genes in the genome [5–7], and plasmids are not stably inherited if selection is not maintained [8]. Thus, it is advantageous and relatively simple to integrate desired genes and whole pathways into the yeast genome, rather than expressing them from a plasmid.

**Competing interests:** The authors have declared that no competing interests exist.

With advances in DNA editing and DNA writing technologies, the scale and complexity of yeast genome engineering is rapidly increasing. This ranges from the creation of synthetic yeast genomes for the Sc2.0 project [9], to their subsequent rearrangement by SCRaMbLE [10–12], to genome-wide targeted perturbations [13, 14]. Thousands of genomically edited strains can be created in a single experiment, necessitating robust high-throughput methods for extracting genomic DNA (gDNA), to then confirm the genotypes by DNA sequencing [15].

Furthermore, to take advantage of DNA-barcode methods that can measure the fitness or phenotype of hundreds of thousands of strain variants [16, 17], extraction methods need to provide high-quality DNA at high efficiency.

Here we describe a fully automated protocol for extraction of gDNA from *S. cerevisiae* that provides both high efficiency extraction and high-quality gDNA. In developing our protocol, we avoided the use of hazardous reagents (e.g., phenol, chloroform), glass beads for mechanical cell disruption [18, 19], or incubation of samples at 100˚C (i.e., boiling) [18], which are difficult to implement with laboratory automation. We show results for DNA quantitation measurements that demonstrate that our protocol can extract gDNA with high efficiency both from cells grown in liquid culture and from colonies grown on agar plates. We also show results from gel electrophoresis that demonstrate that the resulting gDNA is of high quality (i.e., long length and low RNA contamination).

## Results

Our protocol described here is designed to be implemented with an automated liquid handler, though it can also be implemented with manual pipetting following the same steps. Our protocol consists of five major steps:

1. Cell wall and RNA digestion (i.e., spheroplasting) using zymolyase and RNase.

    a. Although this step increases the protocol time when compared with methods using mechanical cell disruption [18], it is important for the quality of the resulting DNA.

    b. In particular, this step preserves intact gDNA with long strand lengths, which is important for the subsequent bead-based purification (step 4) and if long-length PCR products are required (see below).

    c. In addition, this step also removes RNAs. In our experience, without RNase treatment, DNA quantitation measurements can drastically over-estimate the DNA concentration even with fluorescence-based dsDNA methods (e.g., see results obtained with other protocols, below).

2. Cell lysis with SDS. This step simultaneously lyses the spheroplasts and neutralizes proteins.

3. DNA binding and purification by magnetic beads. With the correct proportions of magnetic beads and polyethylene glycol (PEG), only long DNA strands are bound to the beads during this step, and any remaining RNAs or small DNA fragments are removed with the supernatant.

4. DNA washing with 80% ethanol. This step is run twice to completely rinse away PEG, salts, and any other impurities.

5. Elution of gDNA from magnetic beads.

To determine the amount and quality of DNA that can be extracted with our protocol, we first tested with liquid cultures of *S. cerevisiae*. We used cultures grown to mid-exponential growth phase ($OD_{600} = 1.0$) and implemented the automated protocol with 24 samples with a

culture volume of 1.5 mL per sample (arranged in 3 columns of a 96-well plate). We pelleted samples by centrifugation, removed the supernatant (using an automated liquid handler for reproducibility), and stored the resulting cell pellets at -20˚C until use. We ran three replicate tests of the extraction protocol on different days. Across all three replicates, the mean amount of DNA extracted was 1.73 μg with a standard deviation of 0.15 μg (Fig 1; measured using a fluorescence-based dsDNA quantitation method). This is comparable to the theoretical maximum amount of DNA available in the expected number of S. cerevisiae cells at $OD_{600}$ = 1.0 ($3 \times 10^7$ to $4.5 \times 10^7$ cells for a 1.5 mL culture; note that the average number of genomes per cell is > 1 for cells in exponential growth [20]). Within each replicate, samples were arranged and processed in three columns in 96-well plates. The amount of extracted DNA depended slightly on the column (Fig 2). The mean amount of DNA extracted was 1.88 μg, 1.71 μg, and 1.61 μg for columns 1, 2, and 3, respectively. This trend is presumably due to differences in protocol timing for the different columns since an eight-channel head was used in the automated liquid handler (i.e., each column was pipetted at a slightly different time for each step; e.g., the time between addition of the cell lysis buffer and addition of the magnetic beads was 65 s and 130 s longer for columns 2 and 3 respectively than for column 1). The amount of DNA extracted did not depend significantly on the row within each plate (Fig 3).

To assess the quality of the DNA extracted using our protocol, we analyzed the eluted DNA with gel electrophoresis (Figs 4 and 5). When analyzed with a 0.8% agarose gel, for all three

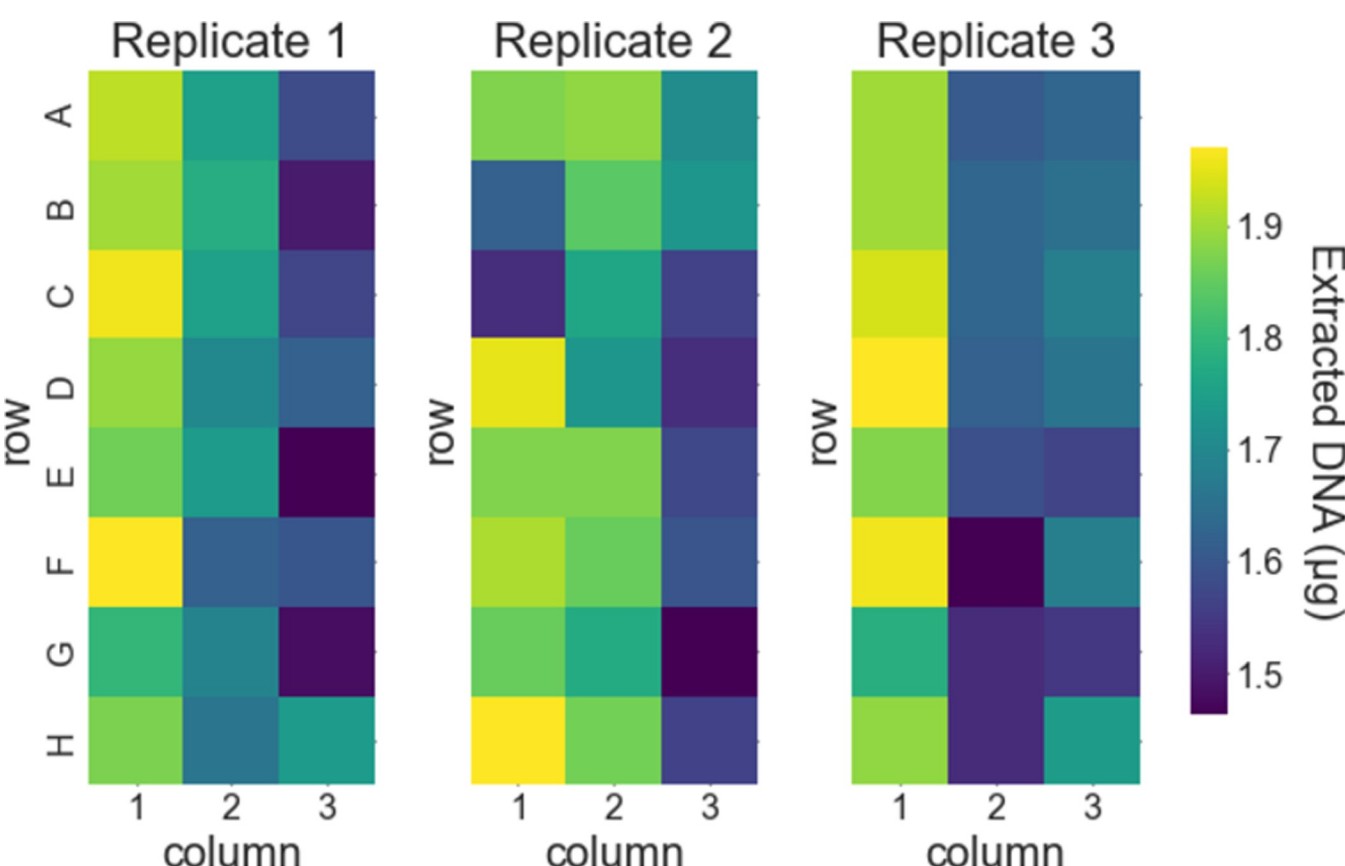

**Fig 1. Amount of DNA extracted from pelleted liquid culture of *S. cerevisiae*.** Results are shown as a heat map for each replicate test of our protocol. The row and column layout of each heat map matches the layout of the samples in the 96-well plate as described in the main text. The final elution volume for the extracted DNA was 100 μL.

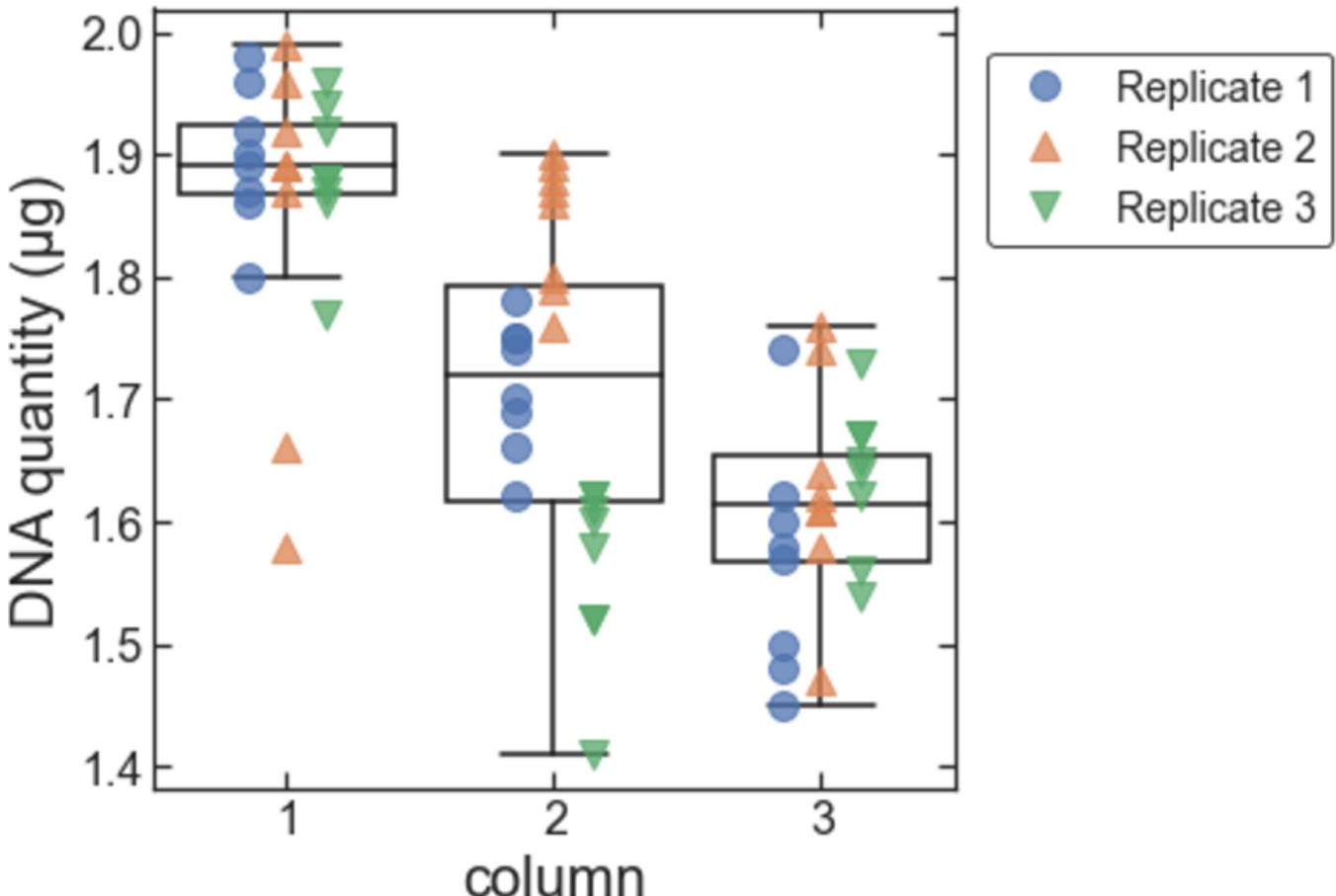

**Fig 2. Amount of DNA extracted from pelleted liquid culture vs. column within the 96-well sample plate.** Data for each replicate are shown as plotted points. The box-and-whiskers plot shows the first, second, and third quartiles for data from each column (box); and the data range excluding outliers (whiskers; outliers are defined as data points less than the first quartile minus 1.5 times the interquartile range).

replicates, all 24 samples showed a dense band on the gel with a length greater than 20 kb (Fig 4). Most of the samples also showed a faint band corresponding to a length of approximately 6 kb. The intensity of the faint band depends on the sample column; it is most evident in samples from column 1 and least evident in samples from column 3. This suggests that the faint band may be contributing to the column-wise trend in the apparent amount of DNA extracted per sample. Based on comparisons with other DNA extraction methods (see below), we think the faint band may be due to a slight amount of residual RNA. To get a more quantitative estimate for the length of the extracted DNA, we also ran one sample on a low-density gel (0.3% agarose). Those results showed a single band with a length of about 50 kb (Fig 5).

UV absorbance ratios, $A_{280}/A_{260}$ and $A_{260}/A_{230}$, are also often used to evaluate DNA extract purity. We used a small-volume spectrophotometer (Nano-Drop) to measure the $A_{280}/A_{260}$ and $A_{260}/A_{230}$ ratios of the DNA extracted with our protocol. The $A_{280}/A_{260}$ ratio was 1.68, which is not significantly below the range typically associated with high purity DNA (1.8 to 2.0) [21, 22]. The $A_{260}/A_{230}$ ratio was 1.01, which is well below the the typical range for high-purity DNA (1.8 to 2.2). Although a low $A_{260}/A_{230}$ ratio is often associated with carryover of phenol or chaotropic salts used in the DNA extraction, those reagents were not used in our protocol. So, the low $A_{260}/A_{230}$ ratio here is probably an indication of protein contamination

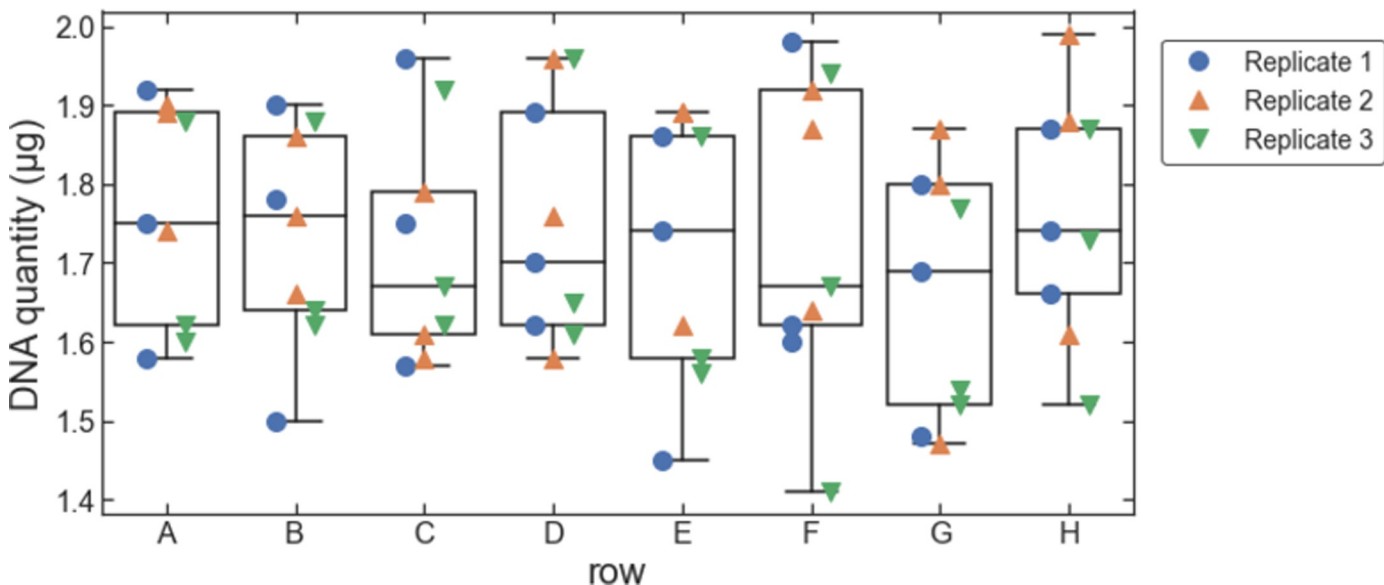

**Fig 3. Amount of DNA extracted from pelleted liquid culture vs. row within the 96-well sample plate.** Data for each replicate are shown as plotted points. The box-and-whiskers plot shows the first, second, and third quartiles for data from each row (box); and the data range (whiskers).

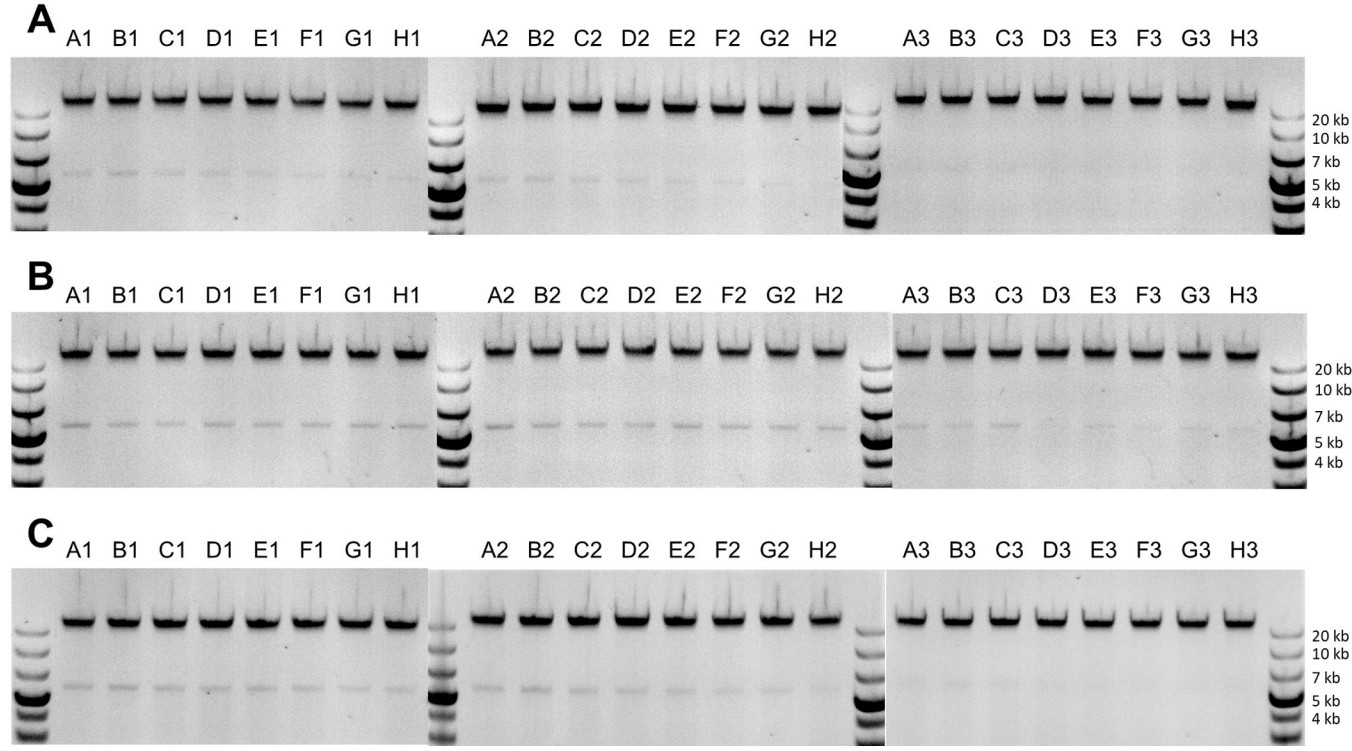

**Fig 4. Gel images for DNA extracted from pelleted liquid culture of *S. cerevisiae*.** (A) Replicate 1. (B) Replicate 2. (C) Replicate 3. The labels above each gel band (e.g., "A1"), indicate the row and column of each extracted sample.

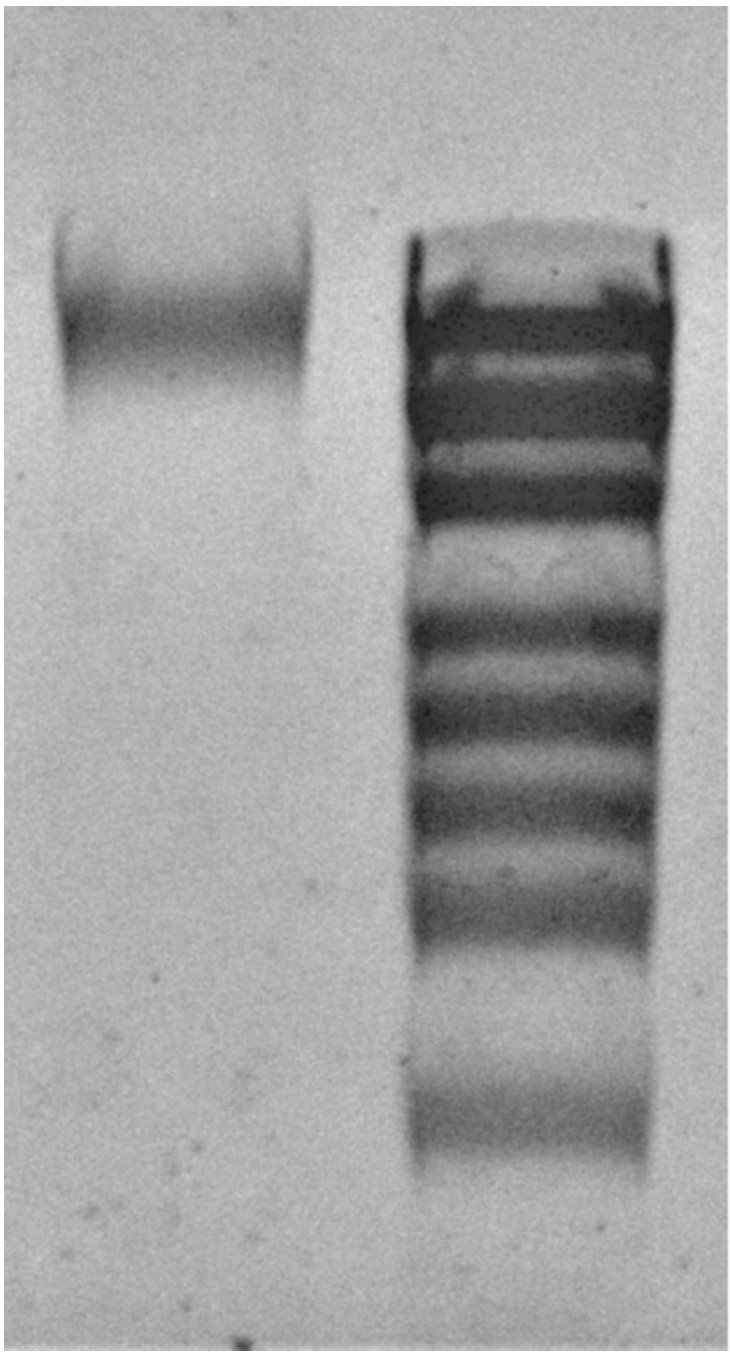

**Fig 5. Low-density gel image for extracted DNA.** DNA extracted from pelleted liquid culture of *S. cerevisiae* is shown on the left. A high-range DNA ladder is shown on the right.

[21]. Although protein contamination was not a problem for our intended down-stream applications (PCR and DNA barcode sequencing), it could be for other applications. For those applications a Proteinase K digestion step can be added after the cell wall and RNA digestion (see optional steps in the protocol, S1 File). In addition, the apparent DNA concentration determined by the small-volume $A_{260}$ measurement was about 1.8-fold higher than the concentration determined using the fluorescence-based dsDNA quantitation method.

To demonstrate the suitability of the extracted DNA for down-stream processing, we used it as template DNA in PCR reactions. We tested two different PCR reactions, with different length products: 1.3 kb and 11.9 kb, and we used gel electrophoresis to analyze the results. Both PCR reactions resulted in a single dense band at the expected length (Fig 6).

Although the main motivation in developing our protocol was to enable automated, high-efficiency gDNA extraction, we also tested its suitability for use with colony PCR. For this, we used the same automated protocol starting with colonies resuspended in 50 μL PBS instead of frozen cell pellets. In our protocol for colony extractions, we reduced the amount of magnetic beads in solution, since a lower DNA binding capacity was needed (see protocol in S1 File and at doi.org/10.17504/protocols.io.8epv592p5g1b/v3.

We ran three replicate tests of our protocol with *S. cerevisiae* colonies. For the first, second, and third replicates, we used small, medium, and large colonies, respectively, of the same *S. cerevisiae* strain. This was to test if the reduced concentration of magnetic beads would work for different sizes of yeast colonies, which we expected to have different amounts of gDNA. The small colonies were grown for 48 hours, and the medium and large colonies were grown for 72 hours. The geometric mean for the amount of DNA extracted was 0.085 μg, 0.28 μg, and 0.43 μg for the first (small colonies), second (medium colonies), and third (large colonies) replicates, respectively (Fig 7). As with the liquid cultures, the amount of extracted DNA depends slightly on the column within the 96-well plate, with samples in column 1 yielding the highest amount and column 3 yielding the lowest amount, on average (Fig 8). And again, as with the liquid cultures, the amount of DNA extracted did not depend significantly on the row within each plate (Fig 9).

To assess the quality of the DNA extracted from *S. cerevisiae* colonies, we analyzed the eluted DNA with gel electrophoresis (Fig 10). Across all three replicates, the most prominent band from all samples corresponded to long gDNA, similar to that seen for extractions from pelleted liquid cultures (> 20 kb). Many of the samples, including all of the samples from medium and large colonies, also resulted in the faint band at approximately 6 kb that was seen

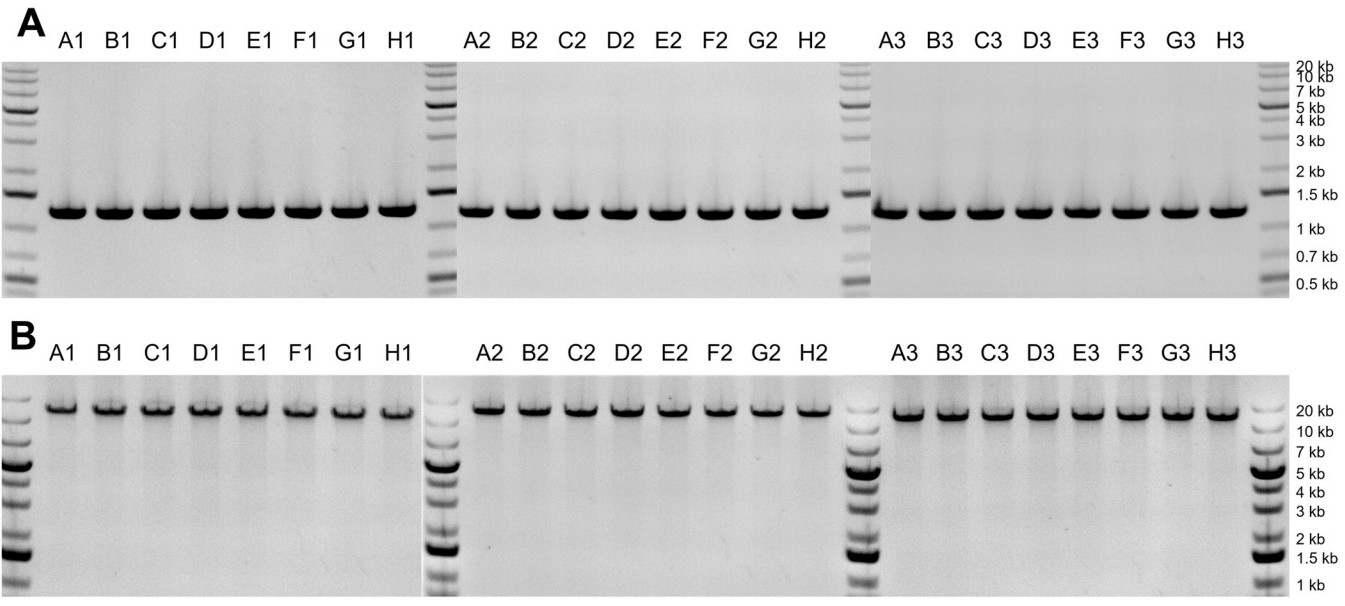

**Fig 6. Gel images for PCR product after extraction from pelleted liquid culture of *S. cerevisiae*.** (A) PCR reaction with expected product length of 1.3 kb. (B) PCR reaction with expected product length of 11.9 kb. The labels above each gel band (e.g., "A1"), indicate the row and column of each extracted sample.

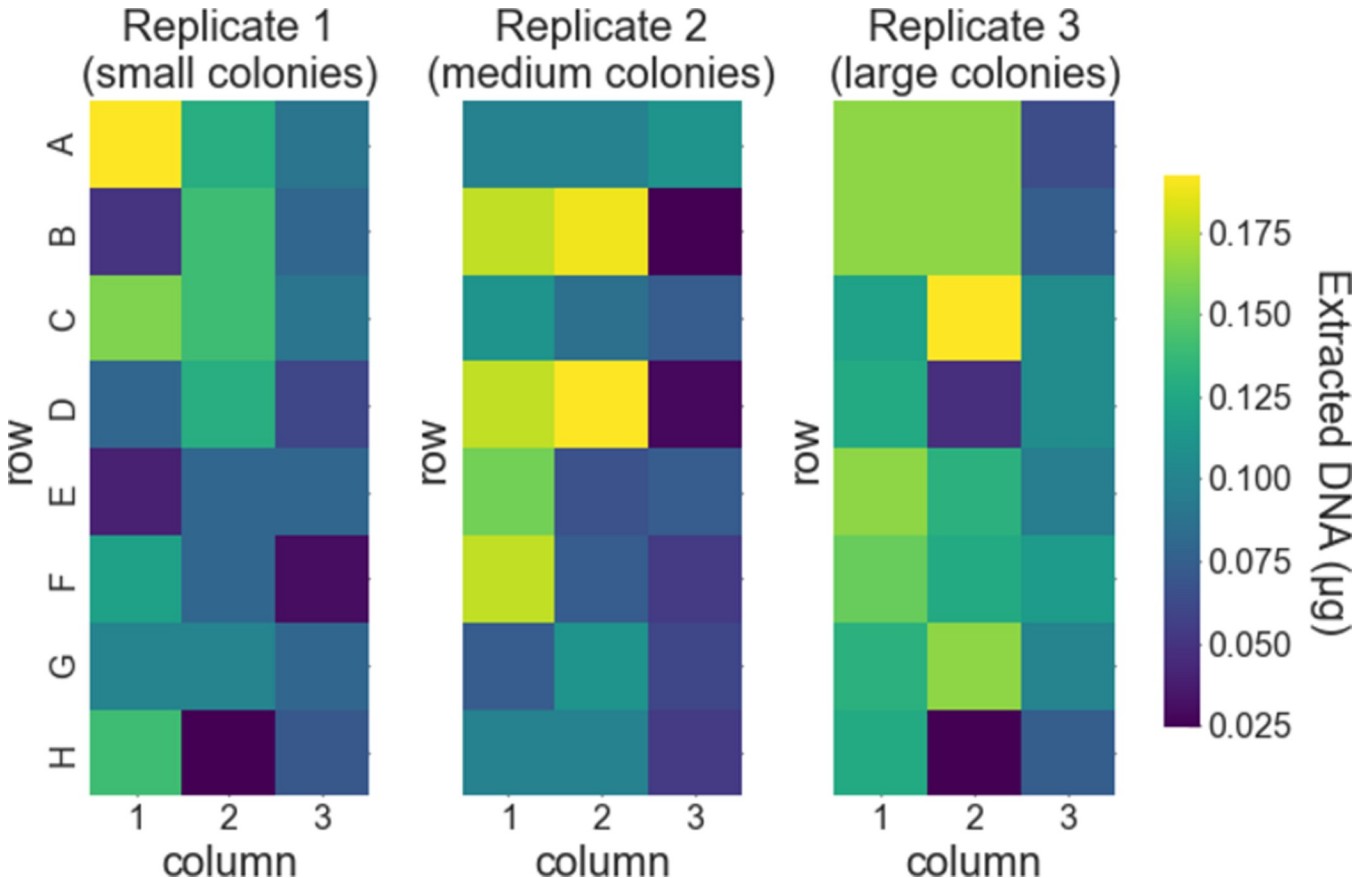

**Fig 7. Amount of DNA extracted from colonies of *S. cerevisiae*.** Results are shown as a heat map for each replicate test of our protocol. The row and column layout of each heat map matches the layout of the samples in the 96-well plate as described in the main text. The final elution volume for the extracted DNA was 100 μL. Replicate 1 was performed with small colonies, replicate 2 with medium colonies, and replicate 3 with large colonies.

with the extractions from pelleted liquid culture. We also used the DNA from colonies as template in PCR reactions and analyzed the resulting PCR product with gel electrophoresis. For all samples, there was a single band at the expected length (1.3 kb; Fig 11).

To compare our automated protocol with existing protocols for yeast gDNA extraction, we also tested several other protocols with frozen cell pellets prepared in the same manner as those used for testing our automated protocol. The apparent amount of DNA extracted was generally comparable to but less than that obtained with our automated protocol, ranging from 0.15 to 1.2 μg of DNA per sample (Table 1). Two of the existing protocols have been described in previous publications with results for the yield of DNA extracted from *S. cerevisiae* cultures. With both of those methods, we obtained lower DNA yields than expected based on the previous publications (assuming that our yeast cultures have between $3 \times 10^7$ and $4.5 \times 10^7$ cells per 1.5 mL): For the fast, high-temperature SDS-based protocol [23], we obtained a total DNA yield of 0.12 μg, compared with an expected yield between 0.5 μg and 0.77 μg ("... yield was... approximately 1.7 μg DNA/$10^8$ cells..."). For the lithium-acetate-based protocol [24], we obtained a total DNA yield of 0.04 μg, compared with an expected yield between 0.3 μg and 0.45 μg ("... yield of 100 ng of gDNA per $1 \times 10^7$ cells..."). The discrepancy between the reported yields and the yields obtained with our testing could be due to differences in culture conditions, though it should be noted that with further optimization, those existing protocols could provide higher DNA yields than what is indicated in Table 1.

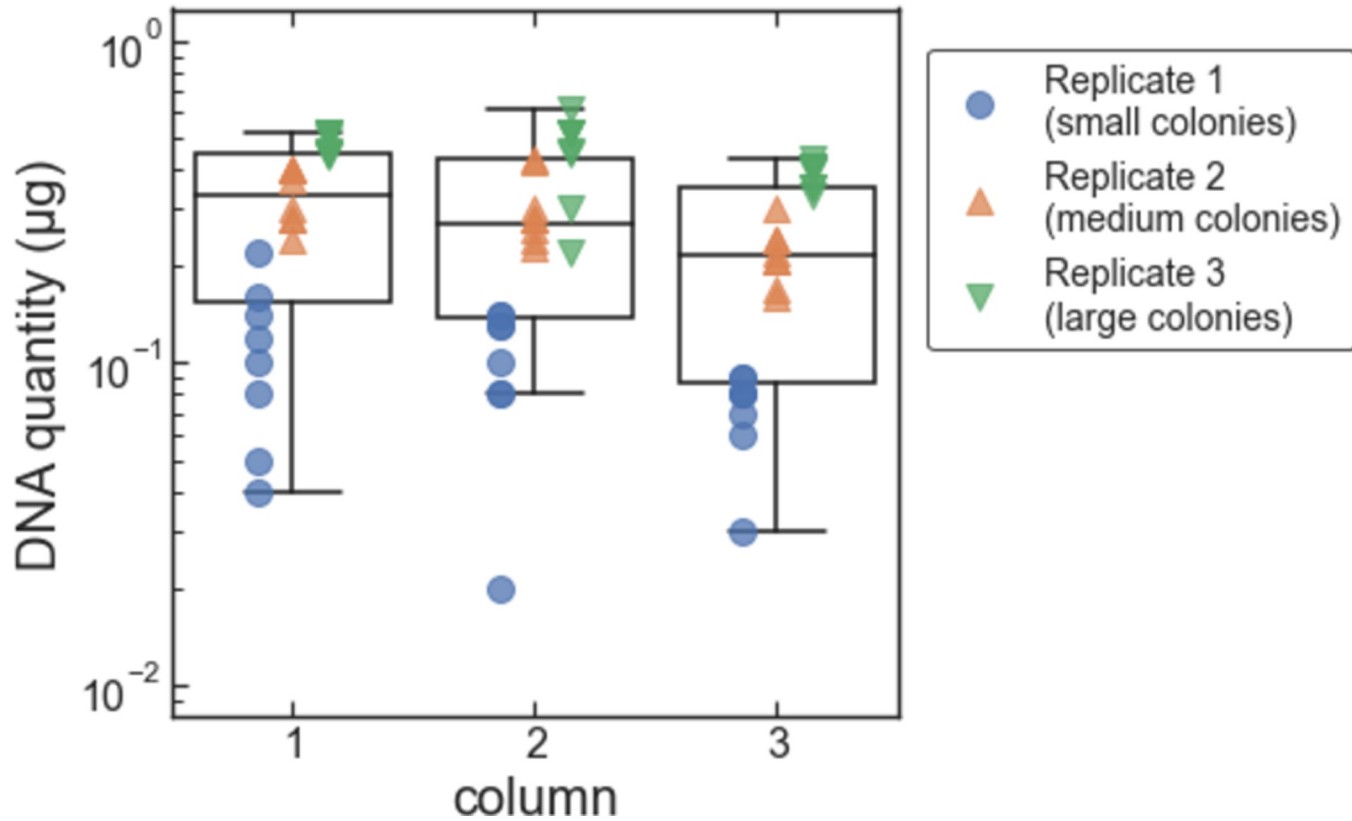

**Fig 8. Amount of DNA extracted from colonies vs. column within the 96-well sample plate.** Data for each replicate are shown as plotted points. The box-and-whiskers plot shows the first, second, and third quartiles for data from each column (box); and the data range excluding outliers (whiskers; outliers are defined as data points less than the first quartile minus 1.5 times the interquartile range).

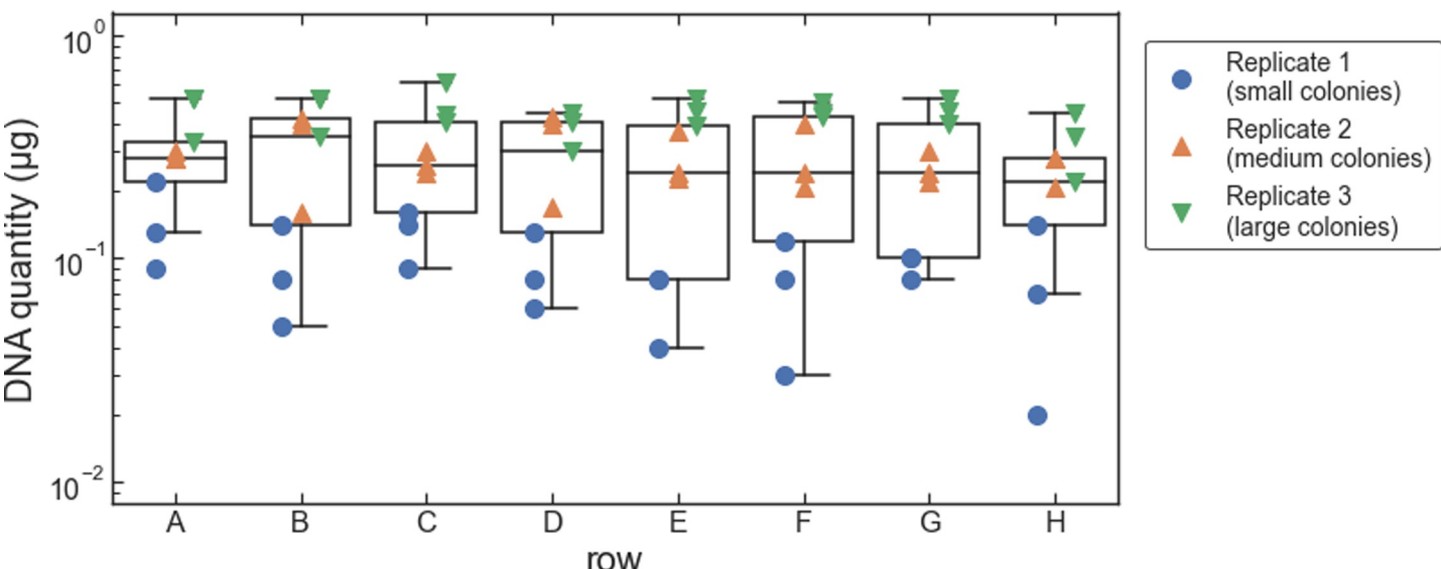

**Fig 9. Amount of DNA extracted from colonies vs. row within the 96-well sample plate.** Data for each replicate are shown as plotted points. The box-and-whiskers plot shows the first, second, and third quartiles for data from each row (box); and the data range excluding outliers (whiskers; outliers are defined as data points less than the first quartile minus 1.5 times the interquartile range).

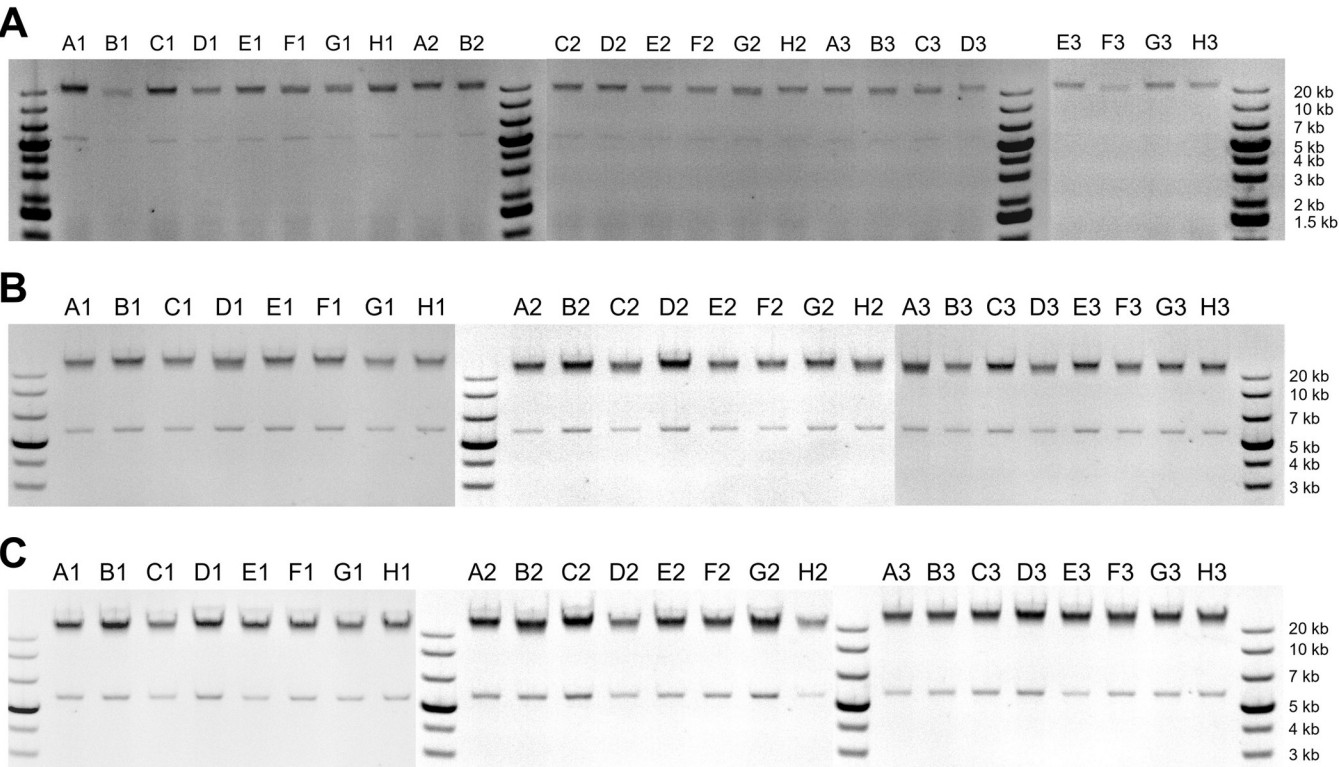

**Fig 10. Gel images for DNA extracted from colonies of *S. cerevisiae*.** (A) Replicate 1 (small colonies). (B) Replicate 2 (medium colonies). (C) Replicate 3 (large colonies). The labels above each gel band (e.g., "A1"), indicate the row and column of each extracted sample.

Additionally, when we analyzed the eluent from some of the other protocols using gel electrophoresis, we found mostly bands corresponding to much shorter lengths, between 0.5 kb and 2 kb (Fig 12A). To understand the source of those short-length bands, we treated each eluent with an extra RNase digestion step (15 min. at 37˚C with a mixture of RNase A and RNase T1) and re-measured the DNA concentration using the fluorescence-based dsDNA quantitation method. The resulting DNA concentration was significantly lower for some of the protocols (Table 1), indicating that the original eluent from those protocols contained large amounts of RNA contamination. Note that although the fluorescent dye used in the dsDNA quantitation method is relatively specific for double-stranded DNA, it can also bind to RNA molecules with secondary structure (e.g., tRNA), and high concentrations of RNA can result in significant overestimation of the true DNA concentration [25]. For the protocols with a difference in the apparent DNA concentration after RNase treatment, we also re-analyzed the eluent with gel electrophoresis (Fig 12B). In the resulting gels, the short-length bands were largely absent, leaving only very faint bands at the expected length (> 20 kb). In contrast, gel electrophoresis of the eluent obtained with our protocol showed similar results before and after the extra RNase treatment (Fig 12, lanes labeled with "1").

## Discussion

The gDNA extraction protocol described here was developed to enable fully automated gDNA extraction and purification for applications that require high-efficiency extractions of high-quality DNA. The total time required for the method is approximately one hour and 45 minutes. This is comparable to the time required for commercial kit protocols but is longer than

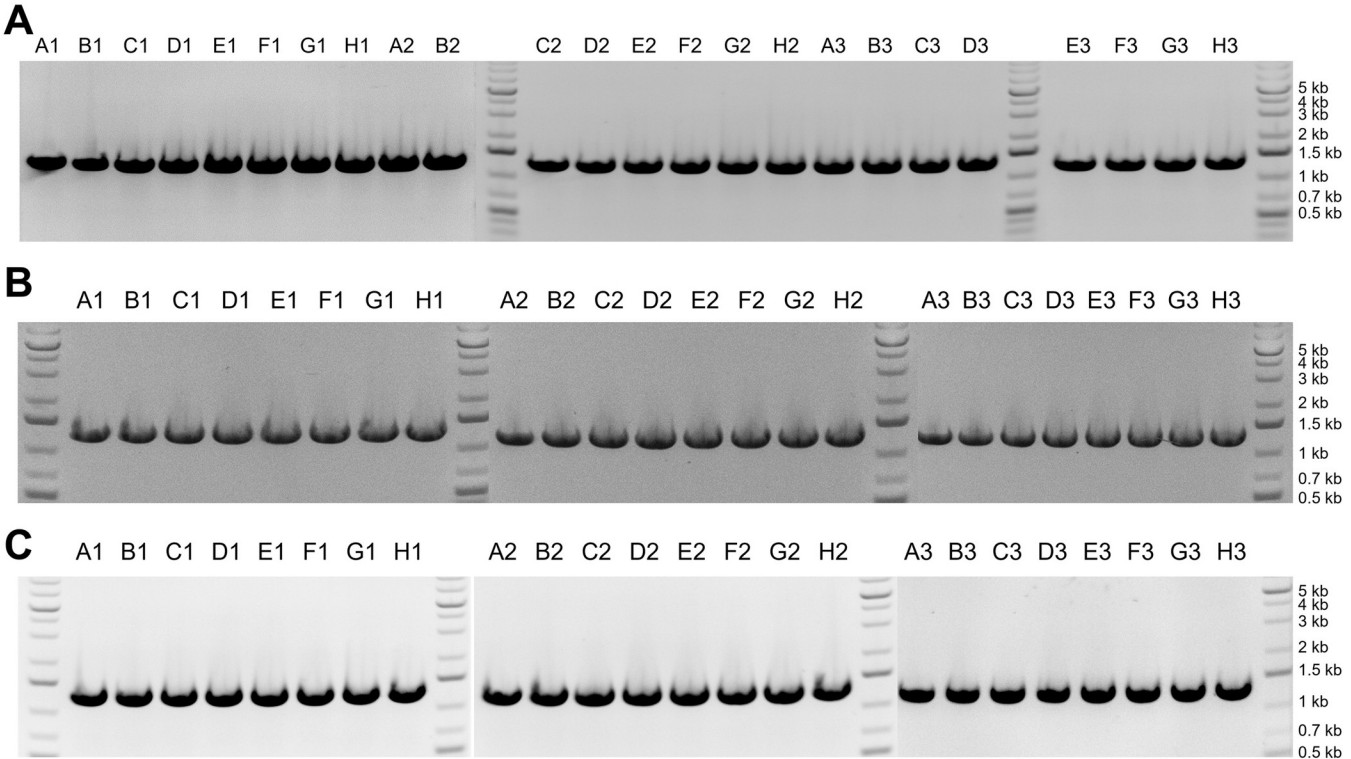

**Fig 11. Gel images for PCR product after extraction from colonies of *S. cerevisiae*.** (A) Replicate 1 (small colonies). (B) Replicate 2 (medium colonies). (C) Replicate 3 (large colonies). The labels above each gel band (e.g., "A1"), indicate the row and column of each extracted sample.

the time for some previously described "fast" or "rapid" protocols [18, 19, 24, 26–28]. Those more rapid protocols generally produce lower amounts of gDNA and/or lower quality gDNA that is suitable for PCR verification of clonal populations but may not be suitable for analysis of large-scale barcoded variant libraries or for PCR amplification of long DNA sequences. As we have shown, our protocol described here produces very long gDNA, and consistently

**Table 1. Comparison of different protocols for genomic DNA extraction from *S. cerevisiae*.**

| | Extraction Protocol | | | | | |
|---|---|---|---|---|---|---|
| | 1 | 2 | 3 | 4 | 5 | 6 |
| Apparent DNA concentration (ng/μL) | 17.3 | 8.7 | 1.9 | 10.5 | 9.7 | 11.5 |
| DNA concentration after RNase treatment (ng/μL) | 17.3 | 8.7 | 1.9 | 1.2 | 0.4 | 3.4 |
| Elution volume (μL) | 100 | 80 | 80 | 100 | 100 | 100 |
| Total DNA (μg) | 1.73 | 0.70 | 0.15 | 0.12 | 0.04 | 0.34 |

The extraction protocols that were tested are:

1. Our new protocol reported here (single sample, implemented with manual pipetting)

2. Commercial zymolyase-based kit with chloroform (single sample)

3. Commercial zymolyase-based kit without chloroform (single sample)

4. Fast, high-temperature SDS-based protocol [23] (two samples, average result shown)

5. Lithium-acetate-based protocol [24] (two samples, average result shown)

6. Commercial magnetic-bead-based kit (two samples, average result shown)

Each protocol was tested using the same type of sample that was used for testing the automated protocol (frozen cell pellets from 1.5 mL of mid-log-phase culture, $OD_{600}$ = 1.0)

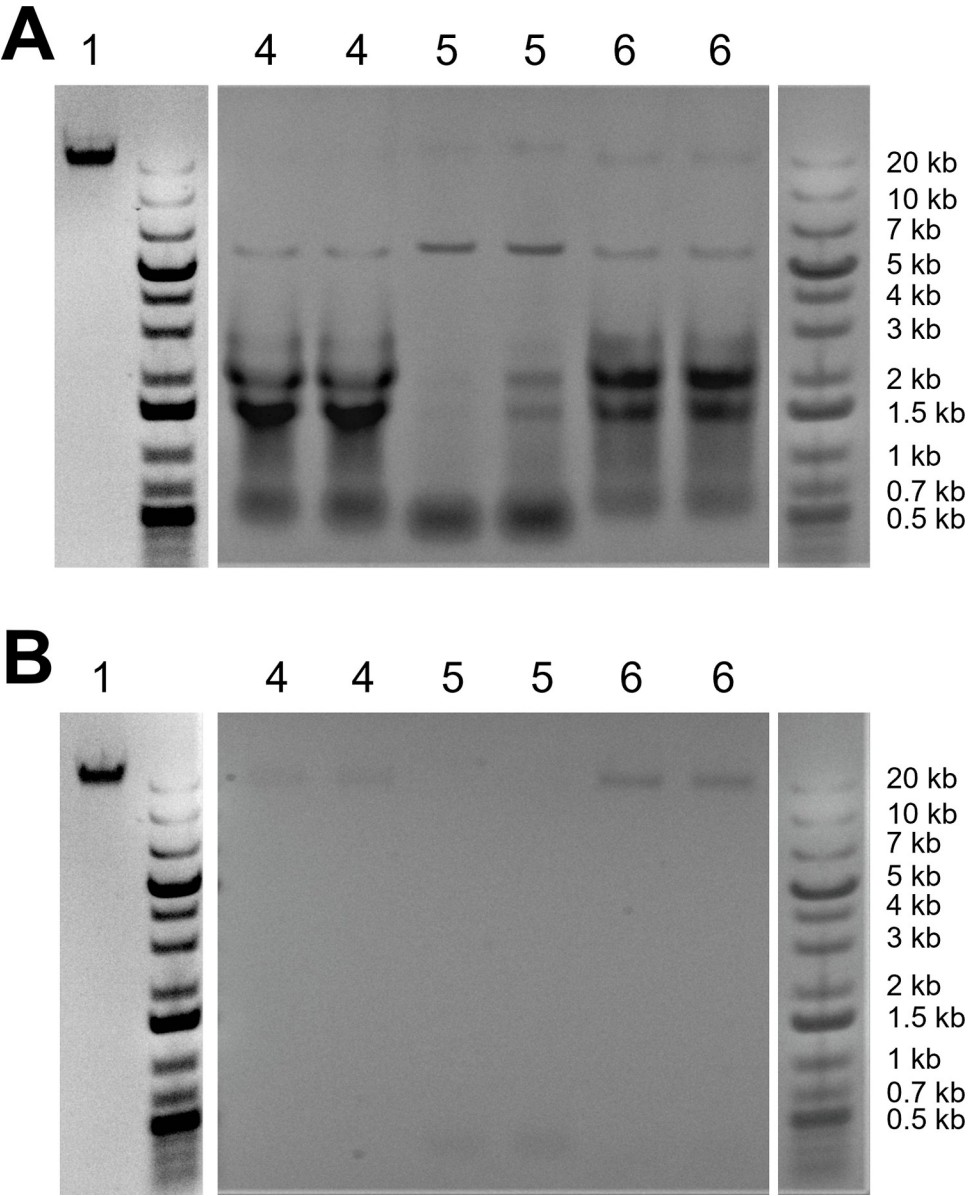

**Fig 12. Gel images comparing DNA extraction protocols.** Gel lanes are numbered to indicate the protocol used for extraction (with the same numbering as in Table 1): 1) our new protocol reported here; 4) fast, high-temperature SDS-based protocol [23]; 5) lithium-acetate-based protocol [24]; 6) Commercial magnetic-bead-based kit. **A.** Gels run with eluent directly from each extraction protocol, before an RNase digestion step. **B.** Gels run with eluant from each protocol after an RNase digestion step (15 min. at 37°C). Note that our new protocol reported here includes RNase digestion, and for that protocol, the difference between (**A**) and (**B**) is in addition of an extra RNase digestion step at the end of the extraction.

provides a high amount of extracted gDNA. These advantages along with the compatibility with high-throughput automated workflows make this protocol ideally suited for future work to measure the population dynamics of barcoded cell variants across multiple growth conditions.

## Materials and methods

Our protocol described in this peer-reviewed article is published on protocols.io (doi.org/10.17504/protocols.io.8epv592p5g1b/v3) and is included for printing as S1 File with this article. Reagents, equipment, and labware required for our protocol are included on protocols.io and in S1 File.

The materials and methods used to analyze the DNA extracted with our protocol, and to compare it with other protocols, are described here.

### Equipment and labware

- Qubit 3.0 fluorometer.

- Applied Biosystems Simply Amp Thermal Cycler.

- Eppendorf Thermo-Mixer C.

- New England Biolabs 12-Tube Magnetic Separation Rack.

- Eppendorf Centrifuge 5804R.

- NIPPON Genetics FAS-Digi PRO Gel Imager.

- PCR plate (Bio-Rad, HSP9645)

- 50 mL Conical Bottom Centrifuge Tubes (FALCON, cat.no. 352098)

- Microcentrifuge Tubes, 1.7 mL, RNase/DNase free (Costar, cat. no. 3207/3620)

### Reagents

- PBS, pH 7.4 (Invitrogen, cat. no. AM9625)

- SDS, 20% (Quality Biological, cat. no. 351-066-101)

- EDTA, 0.5 mol/L, pH 8.0 (Ambion, cat. no.AM9262)

- Lithium Acetate Dihydrate (Sigma-Aldrich, cat. no. L4158)

- YeaStar Genomic DNA Kit (ZymoResearch, cat. no. D2002)

- NucleoMag 96 Tissue KIT (Macherey-Nagel, cat. no. REF 744300.1)

- Ethanol, 200 proof (100%) (Fisher Scientific, cat. # 2716TR)

- Nuclease-Free Water (ThermoFisher Scientific, cat. no. AM9938)

- YPD Agar (BioShop Canada, cat. no. YPD001.500)

- Yeast Nitrogen Base without ammonium sulfate (BioShop Canada, cat. No. YNB404)

- Monosodium Glutamate (BioShop Canada, cat. No. GLU303.500)

- SC Supplement Mixture (Sunrise Science Canada, cat. no. 1300–030)

- Sodium Hydroxide, (Sigma-Aldrich, cat. no. 795429)

- Qubit 1x dsDNA Assay Kit (ThermoFisher Scientific, cat. no. Q33231)

- DNA oligos (Integrated DNA Technologies)

- 2x Phusion Flash High Fidelity PCR Master Mix (Thermo Scientific, cat. no. F548L)

- 2x Phusion Plus PCR Master Mix (Thermo Scientific, cat. no. F631)

- GeneRuler High Range DNA Ladder (Thermo Scientific, cat. no. SM1353)

- GeneRuler 1 kb Plus DNA Ladder (Thermo Scientific, cat. no. SM1333)

- UltraPure Agarose (Invitrogen, cat. no. 16500)

- Invitrogen E-Gel, 0.8% (ThermoFisher Scientific, cat. no. 501808)

- Invitrogen E-Gel, 1.2% (ThermoFisher Scientific, cat. no. 501801)

- TAE Buffer 10x (Quality Biological, cat. no. 351-009-491)

- Midori Green Xtra DNA Stain (NIPPON Genetics, cat. no. MG10)

## Yeast strain construction

The yeast strain used for this study (yBS1039) was constructed using methods and materials described by Shaw et al. [29] and Lee et al. [5]. Strain genotypes are listed in S2 File, and sequences of modified genomic loci are available in S3 File. Plasmids encoding a refactored yeast mating pathway required to link G-protein-coupled receptor (GPCR) activation to a downstream fluorescent reporter gene were constructed and introduced to the genome via CRISPR, following designs published previously [29]. The human cannabinoid receptor CB2 coding sequence was yeast codon optimized and obtained from Genscript, and C-terminally tagged with yeast codon optimized mTagBFP2 [30]. The yeast terminator tDIT1 associated with high protein expression [31] was obtained by PCR amplifying from yeast genomic DNA (Sigma-Aldrich, 69240), with a random barcode added to enable colony-specific studies. The inserted genes and colony-specific barcode were confirmed by PCR of the genomic DNA and Sanger sequencing.

All liquid cultures were grown in SC-COMP+E media consisting of 1.74 g/L Yeast Nitrogen Base without ammonium sulfate, 1 g/L monosodium glutamate, 2 g/L SC Supplement Mixture, 2% glucose, adjusted to pH 5.8 with 1 mol/L NaOH.

Colonies of *S. cerevisiae* strain yBS1039 were prepared by streaking glycerol stock onto YPD plates and incubating for between 2 days and 3 days at 30˚C.

For testing with liquid culture, colonies were picked from YPD plates and mixed into 5 mL of SC-COMP+E media in 50 mL centrifuge tubes. Cultures for each replicate test of our protocol were prepared from separate colonies. Those cultures were incubated for 16 hours at 30˚C shaking at 300 rpm. Each culture was then diluted 9-fold by combining with 40 mL SC-COMP+E media in a 250 mL baffled flask. Each 45 mL culture was incubated at 30˚C shaking at 300 rpm for between 4hours and 5 hours until the optical density ($OD_{600}$) was approximately 1.0. Finally, cultures were aliquoted into a deep-well plate (1.5 mL per well), then pelleted and frozen until use as described in our protocol at proticols.io: doi.org/10.17504/protocols.io. 8epv592p5g1b/v3.

For testing with colonies, YPD plates were incubated for 2 days (small colonies) or for 3 days (medium and large colonies). Each colony was resuspended in 50 μL of PBS in a well in a deep-well plate and used directly with the DNA extraction protocol as described in our protocol at proticols.io: doi.org/10.17504/protocols.io.8epv592p5g1b/v3.

## DNA quantitation

All reported DNA concentrations were determined using a Qubit 3.0 fluorometer with the Qubit 1x dsDNA Assay Kit according to the manual.

## Gel electrophoresis analysis

For the results shown in Figs 4 and 10, 5 μL of extracted DNA was loaded for each sample and run on a 0.8% E-gel. GeneRuler 1 kb Plus DNA ladder was used for size estimation.

For the results shown in Fig 5, 5 μL of extracted DNA was loaded on a 0.3% Agarose gel prepared with TAE buffer and Midori Green Xtra stain (5 μL /100 mL). GeneRuler High Range DNA Ladder was used for size estimation, and the gel was run for 3 hours at 3.3 V/cm.

For the results shown in Figs 6 and 11, 3 μL PCR product was loaded onto the gel for each sample. For analysis of 1.3 kb PCR products (Figs 6A and 11), 1.2% E-gels were used. For analysis of 11.9 kb PCR products (Fig 6B), 0.8% E-gels were used. GeneRuler 1 kb Plus DNA ladder was used for size estimation.

For the results shown in Fig 12, 10 μL of extracted DNA was loaded for each sample and run on a 0.8% E-gel. GeneRuler 1 kb Plus DNA ladder was used for size estimation.

## PCR amplification

For 1.3 kb PCR product (Figs 6A and 11): forward primer: `cattgttctaattattcttattctcctttattctttccta`; reverse primer: `gttgtcgacagtaccttccata`. PCR reactions were prepared with 2x Phusion Flash High Fidelity PCR Master Mix. PCR reactions were run in 96-well plates with 3 μL of each template and 25 μL total volume for each reaction. The thermocycle parameters were:

a. Initial denaturation: 98°C for 60 s

b. 30 cycles:

 i. Denaturation: 98°C for 10 s

 ii. Annealing: 63°C for 20 s

 iii. Elongation: 72°C for 60 s

c. Final Extension: 72°C for 3 min

d. Cooling: 4°C for 60 s

For 11.9 kb PCR product (Fig 6B): forward primer: `gctaaattcgagtgaaacac`; reverse primer: `tctttatatttacatgctaaaaatg`. PCR reactions were prepared with 2x Phusion Plus PCR Master Mix. PCR reactions were run in 96-well plates with 4 μL of each template and 40 μL total volume for each reaction. The thermocycle parameters were:

a. Initial denaturation: 98°C for 60 s

b. 35 cycles:

 i. Denaturation: 98°C for 10 s

 ii. Annealing: 53°C for 20 s

 iii. Elongation: 72°C for 6 min

c. Final Extension: 72°C for 5 min

d.  Cooling: 4˚C for 60 s

## Comparison with existing protocols for yeast gDNA extraction

Several other protocols were tested using frozen cell pellets prepared in the same manner as those used for testing our new automated protocol described here (frozen cell pellets from 1.5 mL of mid-log-phase culture, $OD_{600}$ = 1.0). The other protocols tested were (with the same numbering as in Table 1):

2–3) Commercial "YeaStar Genomic DNA Kit" (ZymoResearch), following protocols from the kit manual with and without chloroform. In both cases, cell pellets were incubated in YD Digestion Buffer at 37˚C for 60 minutes.

4) Fast, high-temperature SDS-based protocol [23], consisting of the following steps:

   a.  cell lysis with SDS at 75˚C.

   b.  protein removal by sodium acetate precipitation.

   c.  DNA precipitation with isopropanol.

5) Lithium-acetate-based protocol [24], with reagent volumes adjusted to match the cell pellet volume: 400 µL of 200 mmol/L lithium acetate solution with 1% SDS, 1200 µL 100% ethanol, 1200 µL 70% ethanol, and elution with 100 µL nuclease-free water.

6) Commercial "NucleoMag 96 Tissue KIT" (Macherey-Nagel) following the protocol from the kit manual with incubation in T1 Buffer (SDS) for 1 hour.

To test for RNA contamination, for each protocol, we treated 20 µL of the resulting eluent with 0.5 µL of RNase A/T1 Mix at 37˚C for 15 minutes. We used the Qubit dsDNA Assay Kit to measure the apparent DNA concentration both before and after this RNase treatment. The results are listed in Table 1.

For the protocols with an apparent difference in DNA concentration after RNase treatment, we also analyzed the eluent before and after RNase treatment with gel electrophoresis (Fig 12).

## Supporting information

**S1 File. Protocol from protocols.io in .pdf format.**
(PDF)

**S2 File. Yeast strain genotypes, in .pdf format.**
(PDF)

**S3 File. Sequences of modified genomic loci, in GenBank format.**
(ZIP)

**S4 File. Source data for Figs 1–3 and 7–9, in .csv format.**
(ZIP)

**S1 Raw images. Raw uncropped images for all gel results, in .pdf format.**
(PDF)

## Acknowledgments

We would like to thank Molly Wintenberg, James Bagley, and Elizabeth Strychalski for thoughtful comments on the manuscript. The yWS677 yeast strain, and various plasmids used in the construction of a refactored GPCR signaling pathway, were generously provided by Dr. Tom Ellis and Dr. William Shaw, Imperial College London. The yeast codon optimized sequence of mTagBFP2 was generously provided by Dr. Alan Moses, University of Toronto.

## Author Contributions

**Conceptualization:** Nina Alperovich, Benjamin M. Scott, David Ross.

**Data curation:** David Ross.

**Formal analysis:** David Ross.

**Investigation:** Nina Alperovich, Benjamin M. Scott, David Ross.

**Methodology:** Nina Alperovich, Benjamin M. Scott.

**Resources:** Benjamin M. Scott.

**Software:** David Ross.

**Validation:** Nina Alperovich.

**Writing – original draft:** Nina Alperovich, Benjamin M. Scott, David Ross.

**Writing – review & editing:** Nina Alperovich, Benjamin M. Scott, David Ross.

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
