## [Decision Letter · Decision Letter 0]

10 Jul 2023

PONE-D-23-18395Automation Protocol for High-Efficiency and High-Quality Genomic DNA Extraction from Saccharomyces cerevisiaePLOS ONE

Dear Dr. Ross,

Thank you for submitting your manuscript to PLOS ONE. After careful consideration, we feel that it has merit but does not fully meet PLOS ONE’s publication criteria as it currently stands. Therefore, we invite you to submit a revised version of the manuscript that addresses the points raised during the review process.

We look forward to receiving your revised manuscript.

Kind regards,

Alvaro Galli

Academic Editor

PLOS ONE

3. We note you have provided a protocols.io PDF version of your protocol and/or a protocols.io DOI. When you submit your revision, please provide a PDF version of your protocol as generated by protocols.io (the file will have the protocols.io logo in the upper right corner of the first page) as a Supporting Information file. The filename should be S1_file.pdf, and you should enter “S1 File” into the Description field. Any additional protocols should be numbered S2, S3, and so on. Please also follow the instructions for Supporting Information captions [https://journals.plos.org/plosone/s/supporting-information#loc-captions]. The title in the caption should read: “Step-by-step protocol, also available on protocols.io.”

Please assign your protocol a protocols.io DOI, if you have not already done so, and include the following line in the Materials and Methods section of your manuscript: “The protocol described in this peer-reviewed article is published on protocols.io (https://doi.org/10.17504/protocols.io.[...]) and is included for printing purposes as S1 File.” You should also supply the DOI in the Protocols.io DOI field of the submission form when you submit your revision.

If you have not yet uploaded your protocol to protocols.io, you are invited to use the platform’s protocol entry service [https://www.protocols.io/we-enter-protocols] for doing so, at no charge. Through this service, the team at protocols.io will enter your protocol for you and format it in a way that takes advantage of the platform’s features. When submitting your protocol to the protocol entry service please include the customer code PLOS2022 in the Note field and indicate that your protocol is associated with a PLOS ONE Lab Protocol Submission. You should also include the title and manuscript number of your PLOS ONE submission.

Reviewers' comments:

Reviewer's Responses to Questions

**Comments to the Author**

1. Does the manuscript report a protocol which is of utility to the research community and adds value to the published literature?

Reviewer #1: Yes

Reviewer #2: No

2. Has the protocol been described in sufficient detail?

To answer this question, please click the link to protocols.io in the Materials and Methods section of the manuscript (if a link has been provided) or consult the step-by-step protocol in the Supporting Information files.

The step-by-step protocol should contain sufficient detail for another researcher to be able to reproduce all experiments and analyses

Reviewer #1: Yes

Reviewer #2: Yes

3. Does the protocol describe a validated method?

Reviewer #1: Yes

Reviewer #2: Yes

4. If the manuscript contains new data, have the authors made this data fully available?

Reviewer #1: Yes

Reviewer #2: Yes

**5. Is the article presented in an intelligible fashion and written in standard English?**

Reviewer #1: Yes

Reviewer #2: Yes

6. Review Comments to the Author

Reviewer #1: This is a good, clear, professionally written paper that can be accepted almost in its current form. However, this study needs more discussion on the suitability of the extracted DNA for the NGS technique such as ONT. The lack of such debate is the primary flow of this paper. Authors should fill this gap. To do this, authors need to provide more measures of DNA purity, such as the standard Nanodrop measures (A260/280, A260/230, and Qubit\\Nanodrop yield ratio (ideally should be 2:1)). Strictly speaking, the gel electrophoresis cannot test the quality of the DNA (line 90, p. 3). I do not see any issues linking their method with the cleanup procedure (upon necessity). It is unnecessary changing the protocol, but to suggest a few extra optional steps. It is unclear to the reader how pure the extracted DNA is, and this issue must be fixed as well (relevant suggestions on p. 2, lines 65 - 69 are insufficient for the reader).

Minor - Table 1. (lines 148-150, 153). More references are necessary.

Reviewer #2: The protocol manuscript describes an automated protocol for extracting high quality DNA from the yeast S. cerevisae without using hazardous chemicals, boiling, or mechanical cell disruption. Although the mais objective of the paper is not clear in the Introduction, the manuscript is well written, the protocol’s steps are clearly described, and the results are robust. The conclusion could be greatly improved to discuss other similiar work that has already been published (for example 10.1128/jcm.40.6.2240-2243.2002 and 10.1111/j.1471-8286.2006.01428.x). In fact, this is the weakest point of the paper. What gap does exist in the literature that the present work fulfils, to justify its publication? It should also be mentioned that the protocols used for comparison are either intrinsically bad or were poorly executed because all yielded bad results. This should be explained or thoroughly reviewed.

7. PLOS authors have the option to publish the peer review history of their article (what does this mean?). If published, this will include your full peer review and any attached files.

Reviewer #1: No

Reviewer #2: **Yes: **Alexandre Dias Tavares Costa

---

## [Author Response · Author response to Decision Letter 0]

30 Aug 2023

We thank the reviewers for their feedback on the manuscript. Or point-by-point response to their comments are below.

Review Comments to the Author

Reviewer #1: This is a good, clear, professionally written paper that can be accepted almost in its current form. However, this study needs more discussion on the suitability of the extracted DNA for the NGS technique such as ONT. The lack of such debate is the primary flow of this paper. Authors should fill this gap. To do this, authors need to provide more measures of DNA purity, such as the standard Nanodrop measures (A260/280, A260/230, and Qubit\\Nanodrop yield ratio (ideally should be 2:1)). Strictly speaking, the gel electrophoresis cannot test the quality of the DNA (line 90, p. 3). I do not see any issues linking their method with the cleanup procedure (upon necessity). It is unnecessary changing the protocol, but to suggest a few extra optional steps. It is unclear to the reader how pure the extracted DNA is, and this issue must be fixed as well (relevant suggestions on p. 2, lines 65 - 69 are insufficient for the reader).

There are different aspects to DNA quality that can be important for different down-stream applications. Gel electrophoresis can indicate whether the DNA is fragmented, for example, which is an important quality criterion for some applications. The absorbance ratios are also sometimes used for assessment of DNA purity, as noted by the reviewer. We have added information about the A260A280 and A260/A230 ratios which indicate the presence of protein continuation (as suspected by the reviewer). In addition, we added optional Proteinase K steps to the protocol, and indicated that they should be used for down-stream applications where protein contamination would be problematic.

Regarding the Qubit/Nanodrop ratio, ideally, both methods should give the same answer – since they are supposed to measure the same thing. However, Nanodrop measurements typically indicate a higher concentration than Qubit, particularly for low concentration samples. Though a ratio of Nanodrop:Qubit less than 2 is often considered to be ok. The Nanodrop:Qubit ratio for DNA extracted with our protocol was about 1.8. We have also added that information to the manuscript.

Minor - Table 1. (lines 148-150, 153). More references are necessary.

There are no citations for the commercial zymolyase-based kit or the commercial magnetic-bead-based kit tested in Table 1. We provide the full kit and vendor information in the Materials and Methods.

Reviewer #2: The protocol manuscript describes an automated protocol for extracting high quality DNA from the yeast S. cerevisae without using hazardous chemicals, boiling, or mechanical cell disruption. Although the mais objective of the paper is not clear in the Introduction, the manuscript is well written, the protocol’s steps are clearly described, and the results are robust. The conclusion could be greatly improved to discuss other similiar work that has already been published (for example 10.1128/jcm.40.6.2240-2243.2002 and 10.1111/j.1471-8286.2006.01428.x). In fact, this is the weakest point of the paper. What gap does exist in the literature that the present work fulfils, to justify its publication? 

We appreciate the reviewer pointing out additional citations that might be relevant. The two papers recommended by the reviewer are for very different types of samples than the samples used in our protocol. The first was specific to medical samples and the second to mammalian tissue. Neither of those publications provides any assessment of the quality of the extracted DNA other than PCR assays. In addition, in the first recommended publication, the only description of the protocol is: 

“For DNA extraction with the MagNA Pure LC, the MagNA Pure LC Total Nucleic Acid Isolation kit (Roche Diagnostics) was used… In the automated DNA isolation process, the samples were dissolved and simultaneously stabilized by incubation with buffer containing guanidinium thiocyanate and proteinase K, total nucleic acids were bound to the surface of glass magnetic particles, unbound substances were removed by several washing steps, and purified DNA was eluted with a low-salt buffer.” 

Most of the details, including the composition of the reagents, are not given – presumably because that information is proprietary. So, if those two recommended citations are representative of protocol publications for DNA extraction, then I’d say that one clear gap that our manuscript fills is that we fully describe our protocol, in sufficient detail that other groups could implement it, and we extensively characterize the performance of our protocol – going well beyond the examples recommended by the reviewer or any other similar publications that we’ve been able to find. 

Additional gaps are identified in the manuscript’s introduction: Given the widespread use of Saccharomyces cerevisiae in the field of engineering biology and the increased use of laboratory automation, we think it is important to provide a well-documented and tested automation protocol specific to that organism. 

It should also be mentioned that the protocols used for comparison are either intrinsically bad or were poorly executed because all yielded bad results. This should be explained or thoroughly reviewed.

We disagree with the reviewer’s opinion on the protocols used for comparison. When we asked other people in the yeast engineering biology community for recommended gDNA extraction protocols, they recommended the protocols that we used for comparison in this manuscript. And although those protocols are good enough for applications that require something simple like a short PCR product from a clonal population, they were not good enough for our applications that require high efficiency and low RNA contamination. Furthermore, although we got lower DNA yields from the comparison methods than what was reported in the original papers on those methods, that discrepancy could be due to differences in the culture conditions (which are not typically described in any detail). 

To address the reviewer’s request here, we have added a short comparison of the results we obtained with the comparison methods and the expected results based on yields reported in the papers describing those methods.

---

## [Decision Letter · Decision Letter 1]

14 Sep 2023

PONE-D-23-18395R1Automation Protocol for High-Efficiency and High-Quality Genomic DNA Extraction from Saccharomyces cerevisiaePLOS ONE

Dear Dr. Ross,

Thank you for submitting your manuscript to PLOS ONE. After careful consideration, we feel that it has merit but does not fully meet PLOS ONE’s publication criteria as it currently stands. Therefore, we invite you to submit a revised version of the manuscript that addresses the points raised during the review process. One reviewer recommends minor language editing.

We look forward to receiving your revised manuscript.

Kind regards,

Alvaro Galli

Academic Editor

PLOS ONE

Journal Requirements:

Reviewers' comments:

Reviewer's Responses to Questions

**Comments to the Author**

1. Does the manuscript report a protocol which is of utility to the research community and adds value to the published literature?

Reviewer #1: Yes

Reviewer #3: Yes

2. Has the protocol been described in sufficient detail?

To answer this question, please click the link to protocols.io in the Materials and Methods section of the manuscript (if a link has been provided) or consult the step-by-step protocol in the Supporting Information files.

The step-by-step protocol should contain sufficient detail for another researcher to be able to reproduce all experiments and analyses.

Reviewer #1: Yes

Reviewer #3: Yes

3. Does the protocol describe a validated method?

Reviewer #1: Yes

Reviewer #3: Yes

4. If the manuscript contains new data, have the authors made this data fully available?

Reviewer #1: Yes

Reviewer #3: Yes

**5. Is the article presented in an intelligible fashion and written in standard English?**

Reviewer #1: Yes

Reviewer #3: Yes

6. Review Comments to the Author

Reviewer #1: Ms can be accepted in its current form. I am satisfied with all responses. All results look good and promising.

Reviewer #3: the paper needs minor languge editing.

the paper has been written in first person language using words like I, we, us or like words. the paper should be written in third persons language.

some sentences are not properly written e.g. the first sentence of the paper

Baker’s yeast (Saccharomyces cerevisiae) is one of the most commonly used organisms for synthetic

biology

I could not understand this sentence properly.

Does the authors mean

Baker’s yeast (Saccharomyces cerevisiae) is one of the most commonly used organisms for for a wide array of biotechnology applications in synthetic biology? or else

there is another sentence in the first paragraph ".................there is greater variability in gene expression from plasmids versus from genes in the genome" what does this mean ? Does the author compare something?

rewriting or restructuring of the following sentences needed for better understanding

"Furthermore, to take advantage of DNA-barcode methods that can measure the fitness or phenotype of

hundreds of thousands of strain variants [16, 17], extraction methods need to provide high-quality DNA

at high efficiency."

Our protocol described here is designed to be implemented with an automated liquid handler, though it

can also be implemented with manual pipetting following the same steps.

Many sentences start with "To" they should be restructured.

"Although the main motivation in developing our protocol..................." motivation cold be replaced by aim or objective or focus.

the language of the paper should be revised throughout with the help of a native English speaker. the paper is not suitable for publication in present form

7. PLOS authors have the option to publish the peer review history of their article (what does this mean?). If published, this will include your full peer review and any attached files.

Reviewer #1: No

Reviewer #3: **Yes: **the review was done by me

---

## [Author Response · Author response to Decision Letter 1]

18 Sep 2023

After consulting with the PLOS ONE Academic Editor, we are resubmitting without a rebuttal.

---

## [Editor Report · Decision Letter 2]

19 Sep 2023

Automation Protocol for High-Efficiency and High-Quality Genomic DNA Extraction from Saccharomyces cerevisiae

PONE-D-23-18395R2

Dear Dr. Ross,

We’re pleased to inform you that your manuscript has been judged scientifically suitable for publication and will be formally accepted for publication once it meets all outstanding technical requirements.

Kind regards,

Alvaro Galli

Academic Editor

PLOS ONE
---

## [Editor Report · Acceptance letter]

6 Oct 2023

PONE-D-23-18395R2 

Automation Protocol for High-Efficiency and High-Quality Genomic DNA Extraction from *Saccharomyces cerevisiae*

Dear Dr. Ross:

I'm pleased to inform you that your manuscript has been deemed suitable for publication in PLOS ONE. Congratulations! Your manuscript is now with our production department. 

Kind regards, 

on behalf of

Dr. Alvaro Galli 

Academic Editor

PLOS ONE